



# loopUI-0.1: uncertainty indicators to support needs and practices in 3D geological modelling uncertainty quantification

Guillaume Pirot[1,2], Ranee Joshi[1,2], Jérémie Giraud[1,2,3], Mark Douglas Lindsay[1,2,4,5], and Mark Walter Jessell[1,2,4]

[1]The Centre for Exploration Targeting, School of Earth Sciences, The University of Western Australia, Perth, Australia
[2]Mineral Exploration Cooperative Research Centre (MinEx CRC), School of Earth Sciences, University of Western Australia, Perth, Australia
[3]GeoRessources Lab, University of Lorraine, Nancy, France
[4]ARC Industrial Transformation and Training Centre in Data Analytics for Resources and the Environment (DARE), Sydney, Australia
[5]CSIRO Mineral Resources, Perth, Australia

**Correspondence:** guillaume.pirot@uwa.edu.au

**Abstract.** To support the needs of practitioners regarding 3D geological modelling and uncertainty quantification in the field, in particular from the mining industry, we propose a *Python* package called **loopUI-0.1** that provides a set of local and global indicators to measure uncertainty and features dissimilarities among an ensemble of voxet models. Results are presented of a survey launched among practitioners in the mineral industry, enquiring about their modelling and uncertainty quantification

practice and needs. It reveals that practitioners acknowledge the importance of uncertainty quantification even if they do not perform it. Four main factors preventing practitioners to perform uncertainty quantification were identified: lack of data uncertainty quantification, (computing) time requirement to generate one model, poor tracking of assumptions and interpretations, relative complexity of uncertainty quantification. The paper reviews and proposes solutions to alleviate these issues. Elements of an answer to these problems are already provided in the special issue hosting this paper and more are expected to come.

## 1 Introduction

One objective of researchers who develop open-source 3D geological modelling algorithms (Loop, 2019; de la Varga et al., 2019) is to make them Findable, Accessible, Interoperable and Reusable (FAIR) for practitioners. As for any software, these algorithms should satisfy the needs and expectations of users (Franke and Von Hippel, 2003; Kujala, 2008). Thus, new developments should rely on a good understanding of modelling purposes, processes and limitations, following a philosophy of

continuous improvement. In a general context, this becomes even more important given the increasing number of open-source algorithms in the fields of earth and planetary sciences (see Figure 1). However, to the best of our knowledge, the needs and uses of 3D geological modelling practitioners with respect to uncertainty quantification are only partially described in the literature, as it constitutes an emerging field that only recently gained traction in both academia and industry.

An essential purpose of modelling is to support decision makers by offering a simplified representation of nature that also

provides a corresponding quantitative assessment of uncertainty, communicating what we know, what remains unknown and

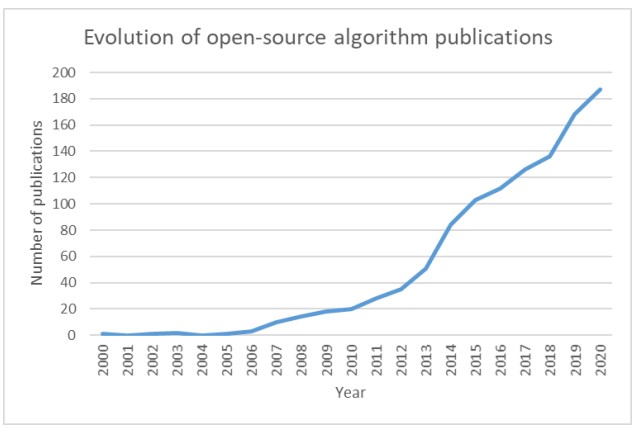

**Figure 1.** Evolution of open-source algorithm publications between 2000 and 2020; data from Web Of Knowledge.

what is ambiguous (Ferré, 2017). Uncertainty quantification is essential, because it allows us to mitigate predictive uncertainty (Jessell et al., 2018) by expanding our knowledge, rejecting hypotheses (Wilcox, 2011) or falsifying scenarios (Raftery, 1993). Questions related to 3D geological modelling and uncertainty quantification are not just limited to the minerals industry, but also concern the fields of $CO_2$ sequestration (Mo et al., 2019), petroleum (Scheidt et al., 2009) and geothermal (Witter et al.,

2019) energy resources as well as hydrogeology (Linde et al., 2017) or civil engineering in urban environments (Osenbrück et al., 2007; Tubau et al., 2017). Here, we are interested in the uses and practices of the minerals industry, that is dealing with both sedimentary basin and hard-rock and/or cratonic settings across regional to mine scales.

The three main pillars of uncertainty quantification are the characterization of uncertainty sources, their propagation and mitigation throughout the modelling workflow (see Figure 2). The different sources of uncertainty, often overlooked, are re-

lated to measurement errors, interpretations, assumptions, modelling approximations and limited knowledge (sample size or unknown process). Measurement or data errors can be estimated by repetitive sampling or from instrument characteristics; they can be propagated through the modelling workflow by Monte Carlo data perturbation (Wellmann and Regenauer-Lieb, 2012; Lindsay et al., 2012; Pakyuz-Charrier et al., 2018). Combined with expert knowledge, initial dataset can be used to shape some assumptions and define plausible conceptual models; but despite the importance of conceptual uncertainty on predictions (Pirot

et al., 2015), it is too often limited to the definition of a unique scenario (Ferré, 2017). From a perspective on algorithms, some assumptions such as how to set parameter ranges are needed and this can greatly impact the definition of geological parameters (Lajaunie et al., 1997) prior to running predictive numerical simulations. Another aspect that is not always considered is the uncertainty related to a spatially limited sampling. Unsampled locations suggest a high uncertainty about the spatial distribution of the model parameters (or values of the property field); this is why it is preferable to resort to spatial stochastic simulations

(e.g. Sequential Gaussian Simulations in a multi-Gaussian world) rather than interpolations (e.g. kriging) to generate models that are parameter fields (Journel and Huijbregts, 1976).

While the main objective of uncertainty quantification and data integration might be to improve the confidence level of predictions for decision making, it usually involves the generation of model ensembles via Monte Carlo algorithm and it is



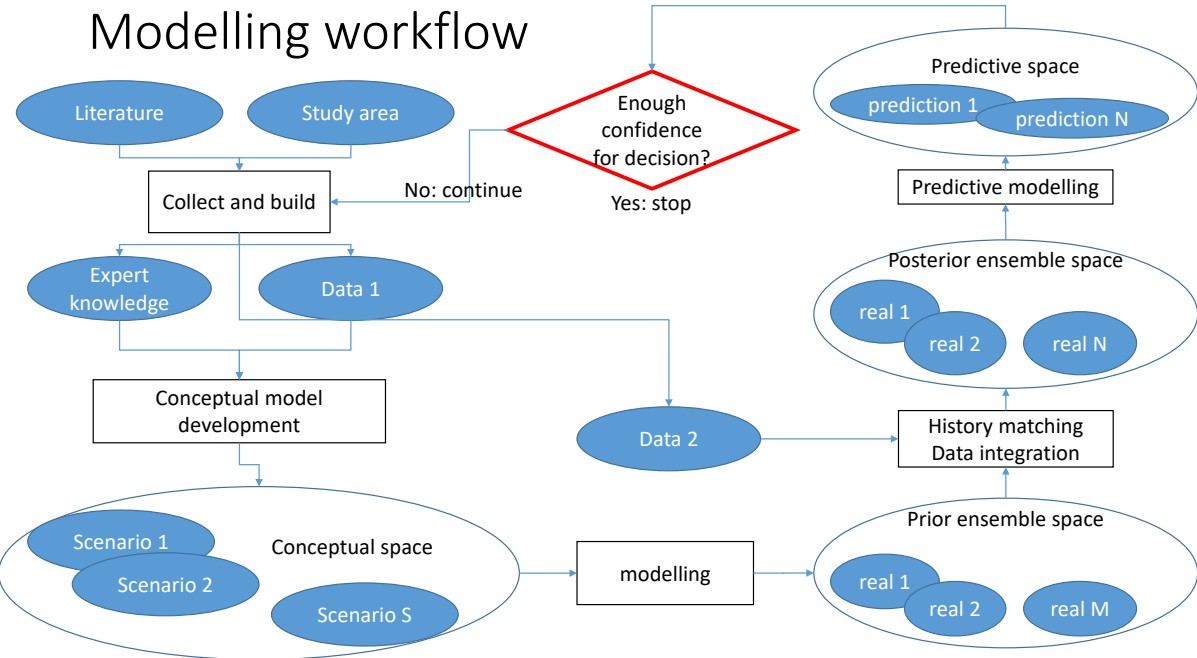

**Figure 2.** Schematic representation of a geo-modelling workflow; each ellipse is associated with some uncertainty.

rarely a straightforward step. Indeed, because of the high dimensionality and non-linearity of earth processes or the lack of
data, history matching might prove difficult to achieve and predicted outcomes might present multiple modes (e.g. **?**Sambridge,
2014; Pirot et al., 2017). In such cases, geological uncertainty analysis allows to improve our understanding of geological
model (dis)similarities and how specific or shared features can be related to upstream parameters and downstream predictions.
Local uncertainty indicators such as voxel-wise entropy or cardinality (Lindsay et al., 2012), computed over an ensemble of
geological voxels will inform about property field variability at specific locations (voxels) of the model mesh. Global indicators
or summary metrics might be useful to identify how the statistics of specific patterns (e.g. fault or fracture network density,
anisotropy, connectivity, etc.) evolve between different models and might also be a way to perform model or scenario selection
(e.g. Pirot et al., 2019) or to reduce the dimensionality of the sampling space, from a high dimensional geological space to
a low dimensional latent space (Lochbühler et al., 2013). Though some indicators have been used or developed for specific
studies or softwares (Li et al., 2014), to the best of our knowledge, no independent uncertainty analysis tool applicable to both
discrete and continuous property fields, combining local and global indicators is available to practitioners.

To investigate the uses and practices of the minerals industry regarding modelling and uncertainty quantification, we re-
cently conducted a survey among numerical modelling practitioners from industry, government and academia in the sector of
exploration and production of economic minerals. In this paper, first, we present the main results and interpretations from this
survey, which questions are listed in Appendix A. Second, to answer these needs, we propose a set of indicators to quantify
geological uncertainty over an ensemble of geological models characterized by lithological units and their underlying scalar-





field derived from implicit modelling. The various indicators are illustrated with a synthetic case derived from a simplified Precambrian dataset of the Hamersley, Western Australia. The *Python* code called **loopUI-0.1** (Pirot, 2021) and the notebooks used to compute and illustrate these indicators are available at https://doi.org/10.5281/zenodo.5656151. 5656151

## 2  Survey

### 2.1  Material and method

The survey was designed to be concise to encourage participation but with several open-ended questions to maximize our chance to learn about different uses and practices, as well as to minimize induced bias whenever possible. The survey is in two parts. Its first part was general and enquired about the scales and dimensions (questions 1 and 2) of geological models. The second part of the survey was more specific at a fixed modelling scale. It enquired about outputs or objectives (questions 3 and 4), about data input (question 5), current workflows (questions 6 to 9) and limitations and expected improvements (question 10). It was distributed between October 2019 and January 2020 among the 3D Interest Group (3DIG), Centre for Exploration Targeting (CET) members, Loop researcher and related networks. Respondents had the opportunity to complete the survey on paper, in a text file or online. A total of 35 responses were collected and anonymised. Seven responses concerned models at different scales, and when the second part of the survey was not clearly duplicated for each scale, the answers were considered with caution for each modelling scale.

### 2.2  Results

This section summarises the answers provided by the survey respondents. Table 1 provides the general statistics of answers to the survey. The high answer rate (mostly over 80%) indicates that questions are meaningful for the respondents and indicates that the reader can have confidence in the presented results. The most answered question is Q1, about the modelling scale. The least answered question is Q9, about data integration and upscaling . The second least answered question is Q8, about the modelling workflow used. All other questions have an answer rate above 80% on a global average. Note that global number of survey answers is smaller than the sum of each survey answers grouped by scale, as some survey answers were returned only once for multiple scales.

Due to existing overlap of collected answers on different questions, the results presented hereafter summarize and group the collected answers by theme (Output or objectives, Input data, Current modelling workflow and Limitations), as outlined in the survey (see Appendix). Each theme is treated by modelling scale (see Figure 3) when it involves different answers.

Q3-4. The main modelling objectives depend on the scale of investigation. At the largest scales, investigation and modelling are particularly useful to assist exploration and prospectivity mapping. At the Greenfields or Regional scale (>10km dimension, ˜ 1km resolution) , the main objective is to obtain a contextual and conceptual understanding of the regional geology, in particular regarding stratigraphy, topology and geochronology. At the Brownfields scale (1-10km dimension, ˜ 100m resolution), modelling aims to estimate deep structure beyond the depths reached by drilling, such as depth of a particular interface or the





**Table 1.** Global number and rate of answers per question and detailed by modelling scale.

|  |  | Total | Q1 | Q2 | Q3 | Q4 | Q5 | Q6 | Q7 | Q8 | Q9 | Q10 |
|---|---|---|---|---|---|---|---|---|---|---|---|---|
| **All scales** | Answer rate |  | 100% | 97% | 94% | 83% | 91% | 89% | 86% | 77% | 60% | 83% |
|  | Nb. answers | 35 | 35 | 34 | 33 | 29 | 32 | 31 | 30 | 27 | 21 | 29 |
| **Mine scale** | Answer rate |  | 100% | 93% | 100% | 71% | 93% | 86% | 86% | 64% | 43% | 79% |
|  | Nb. answers | 14 | 14 | 13 | 14 | 10 | 13 | 12 | 12 | 9 | 6 | 11 |
| **Brownfields scale** | Answer rate |  | 100% | 94% | 100% | 82% | 94% | 94% | 94% | 82% | 76% | 82% |
|  | Nb. answers | 17 | 17 | 16 | 17 | 14 | 16 | 16 | 16 | 14 | 13 | 14 |
| **Greenfields scale** | Answer rate |  | 100% | 100% | 88% | 75% | 81% | 81% | 88% | 81% | 63% | 81% |
|  | Nb. answers | 16 | 16 | 16 | 14 | 12 | 13 | 13 | 14 | 13 | 10 | 13 |

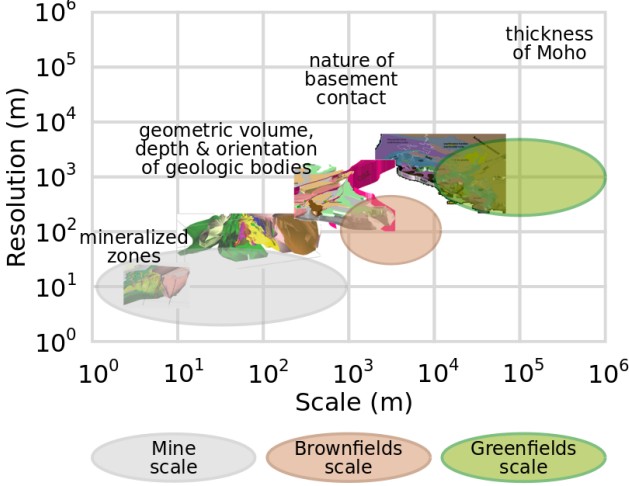

**Figure 3.** Modelling scales and resolutions in the mineral industry with examples of scale-specific scientific enquiry; model illustrations of the Vihanti-Pyhäsalmi Area, Finland, adapted from Laine et al. (2015).

delineation of potential mineral system components (e.g. fluid-alteration pathways and ore deposition environments). At the Mine scale (<1km dimension, ˜ 10m resolution), the objectives are to assist with near mine exploration, resource estimation, ore body localization, drill targeting, operation scheduling and efficient mining, by characterizing local structures and geometries of stratigraphy and mineralization. In addition to fulfilling these various objectives, 3D models are useful to improve hydrogeological characterization, to identify critical zones where more knowledge has to be gained , to allow for comparison between datasets and test internal consistency, scenarios and last but not least, for visualization and communication.



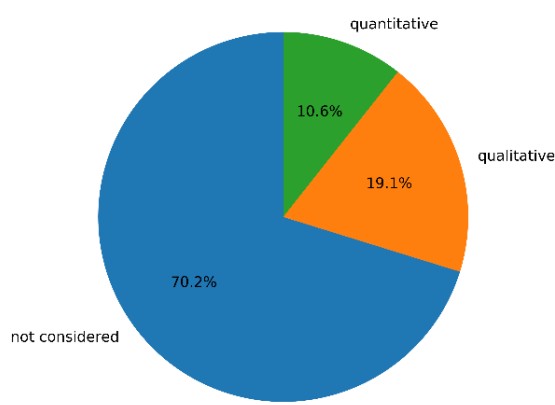

**Figure 4.** Consideration of input data uncertainty in modelling.

Q5. Data types used as input for geoscientific modelling are similar and complementary across scales. For instance, geological mapping, geophysical data (gravity, magnetics, electro-magnetics , seismic), and geochemistry are useful at all scales even
though the related measurements might inform about a regional trend. Drillhole-derived data is usually much more abundant at the Mine scale and can be useful to understand the regional context if analysed with this alternative use in mind. For example, drill logs that only record the presence or absence of mineralisation are not useful for the regional context, however those that record rock properties, lithology and structure can be extrapolated to larger scales and integrated in regional models (e.g. Lindsay et al., 2020).

Q6. Regarding current modelling practices, although the survey respondents acknowledge the importance of uncertainty quantification and propagation into modelling to estimate the confidence around predictions, only about 10% perform quantitative uncertainty characterization and propagation from input data; about 20% do some qualitative characterization and a vast majority of 70% recognize that it is ignored (see Figure 4).

Q7-9. Modelling steps, assumptions or geological interpretations are not recorded in one third of the cases. In most cases,
assumptions and elements of the modelling process are described in metadata or in separate reports. In very few cases, specific procedures and tools are available to keep track of those. The respondents use a variety of software, platforms or programming tools to produce 3D geological models and further data-integration. The lack of a coherent workflow introduces the potential for uncertainty, error and loss of precision when forced to translate data formats across multiple software platforms, in addition to the time lost that could otherwise be dedicated to solving the problem under question or exploring alternative hypotheses.

Q10. Overall, the main limitations involved in 3D geological modelling are uncertainty underestimation due to strong constraining and underlying assumptions (e.g. lack of consideration of alternative conceptual models, lack of uncertainty around interpreted horizons, etc.), navigation between scales, the amount of work required to process and prepare input data (including necessary artificial data), the time required to generate one model, the difficulty to integrate 2D data in a 3D framework, the difficulty to visualize and manage a huge amounts of drillhole data in 3D, the advent of geological inconsistencies, workflow





and model reproducibility given the same inputs, joint integration of geological and geophysical data, lack of tools to visualize uncertainty.

## 3 Uncertainty indicators for categorical or continuous property fields

The estimation of prediction confidence in geosciences relies heavily on numerical simulations and requires generation of an ensemble of models. As indicated by the survey results, an important aspect of uncertainty quantification is its representation.

It helps identifying specific features or zones of interest. Moreover, respondents estimate that such visualization tools are missing. The uncertainty indicators presented hereafter provide a way to identify zones of greater or smaller uncertainty as well as (dis-)similarities between geo-model realizations.

Geo-models can be used to convey very different discrete or continuous properties. Discrete or categorical properties such as lithological formations or classification codes should be compared carefully. Indeed, while it is straightforward to state if two

values are identical or different, additional information is needed to rank dissimilarities. Continuous properties such as potential fields or physical properties (e.g. porosity, conductivity, etc.) do not present this ambiguity to compare range of values. One can note, that depending on how physical properties are assigned during modelling, their value spectrum might be discrete. Some modelling platforms may produce a discrete physical property value spectrum depending on how physical properties are assigned or what input constraints are enforced during model construction.

Local measures of uncertainty provide indicator voxels of the same dimensions than the voxels of a model ensemble. For a given voxel, uncertainty indicators are computed from the distributions of values taken by a given property at the corresponding voxel (same location) across the ensemble of model realizations. Such local indicators are very convenient for visualization: by sharing the same voxel as the model realizations, it is relatively easy to spot zones of low or high uncertainty. However, to be useful, it requires advanced modelling that integrates some spatial constraints, and possibly computationally expensive,

in particular if they are produced by inversion algorithms. Here, we propose to compute Cardinality and Shannon's Entropy for discrete properties (e.g. Wellmann and Regenauer-Lieb, 2012; Pakyuz-Charrier et al., 2018), and similarly, range, standard deviation and continuous Entropy for continuous properties (e.g. Marsh, 2013; Pirot et al., 2017).

Global measures rely on the computation of summary statistics or on the identification of feature characteristics, independently from their locations. The dissimilarities between summary statistics or characteristics can be estimated via appropriate

metrics such as the Wasserstein distance (Vallender, 1974) or the Jensen-Shannon divergence (Dagan et al., 1997) for instance. The resulting global measures allow comparing models with voxels or meshes of different dimensions. However, the computation of summary statistics might be more time consuming than local measures. Their main advantage is that it allows to focus on pattern similarity between models which might be particularly useful to explore the selection of alternative scenarios or prior realizations, before data integration and history matching. In what follows, we propose a series of dissimilarity mea-

sures, applicable to categorical and continuous property fields, based on one-point statistics (histogram Dagan et al., 1997), two-point (geo)statistics (semi-variogram Matheron, 1963), multiple-point statistics (multiple-point histogram Boisvert et al.,





2010), connectivity (Renard and Allard, 2013), wavelet decomposition (e.g. Scheidt et al., 2015) and topology (Thiele et al., 2016, e.g.).

To illustrate the different indicators, inspired by a Precambrian geological setting, that is a simplified dataset from the
Hamersley region, Western Australia, we generated synthetic ensembles of 10 model sets for three different scenarios. Each model set is composed of a lithocode voxet describing the lithological units (categorical variable), and of its underlying scalar-field voxet (continuous variable). The underlying scalar-field is obtained by composition of the different scalar-fields for each unconformable stratigraphic group. Scenario 1 considers all synthetic input data, while scenario 2 keeps only 50% of the data within a North-South limited band and scenario 3 retains input data with a 50% probability (see Figure 5). For each scenario,
the positions and orientations of the input data are perturbed to provide 10 stochastic realizations. Positions are perturbed with a Gaussian error of zero mean and 3m standard deviation. Orientations are perturbed with a von Mises Fisher error of $\kappa = 150$ (corresponding to about $\pm 5 \deg$ of error). Each 3D model was generated using *LoopStructural*(Grose et al., 2021).

## 3.1   Cardinality

The Cardinality of a set, in Mathematics, is the number of elements of the set. Here, we define the Cardinality for a given voxel
as the number of unique elements over the ensemble of models for the corresponding voxel (Lindsay et al., 2012). Computed for all voxels, it provides a Cardinality voxet of the same dimensions than the model voxets. It assumes that all voxels of the model ensemble have the same dimensions. By definition, this indicator can be compute on discrete or categorical property fields. For continuous property fields, we propose to use the range between the minimum and maximum value, the standard deviation. Eventually, these continuous indicators can be normalized over the voxet and then averaged or weighted to provide
another indicator. Figure 6 shows a Cardinality voxet computed from an ensemble of lithocode voxets as well as the range, standard deviation and their normalized average from an ensemble of density voxets.

## 3.2   Entropy

Shannon's entropy (see Eq. 1) is a specific case of the Rényi entropy when its parameter $\alpha$ converges to 1 (Rényi et al., 1961). It has been applied to geo-models since a few decades already (Journel and Deutsch, 1993; Wellmann and Regenauer-Lieb,
2012) and is a finer way than cardinality to describe uncertainty over an ensemble of models, as it takes into account the histogram proportions of the unique values encountered. Let us consider the categorical or discrete variable $X$ that represent a voxel property and assume that it can take $n$ distinct values among an ensemble of voxels. By denoting the probability of observing the $i^{th}$ possible value as $p_i$, the entropy $H$ of $X$ is computed as follows.

$$H(X) = -\sum_{i=1}^{n} p_i \ln p_i \tag{1}$$

For continuous property fields, one need to discretize the continuous domain and integrate along the width of the bins (Marsh, 2013). Here, for the considered example, we choose 50 regular bins. Figure 7 displays Shannon's entropy for the lithocode voxets and the continuous entropy for the other ensemble of property fields: magnetic field, gravity field, density and magnetic susceptibility.







**Figure 5.** Example model set of lithocode and scalar fields for each of the three scenarios; the left column illustrates input data and model set for scenario 1 when all data is considered; the middle column illustrates input data and model set for scenario 2 when 50% of the data within a band is considered; the right column illustrates input data and model set for scenario 3 where each input data is decimated with a probability of 50%.

### 3.3 Histogram dissimilarity

Given a pair of voxels $V_P$ and $V_Q$, we measure the dissimilarities between their histograms by computing a symmetrized and smooth version of the Kullback-Leibler divergence (Kullback and Leibler, 1951) known as the Jensen-Shannon divergence or total divergence to the average (Dagan et al., 1997). Given two random variables $P$ and $Q$, the Jensen-Shannon divergence is conmputed as $\mathrm{JSD}(P||Q) = \frac{1}{2}\mathrm{KLD}(P||M) + \frac{1}{2}\mathrm{KLD}(Q||M)$, where $M = \frac{P+Q}{2}$ and KLD is the Kullback-Leibler divergence.





**Figure 6.** Horizontal sections of cardinality voxets for a categorical property field (first row) or similar indicators for a continuous property field (second to fourth row); the first row shows the cardinality over the ensemble of lithocode voxets, the second row displays the (max-min range) over the ensemble of scalar-field voxets, the third row presents the standard deviation over the ensemble of scalar-field voxets, the fourth row displays the averaged normalized range and standard deviation over the ensemble of scalar-field voxets; the left column refers to scenario 1, the middle column to scenario 2 and the right column to scenario 3.

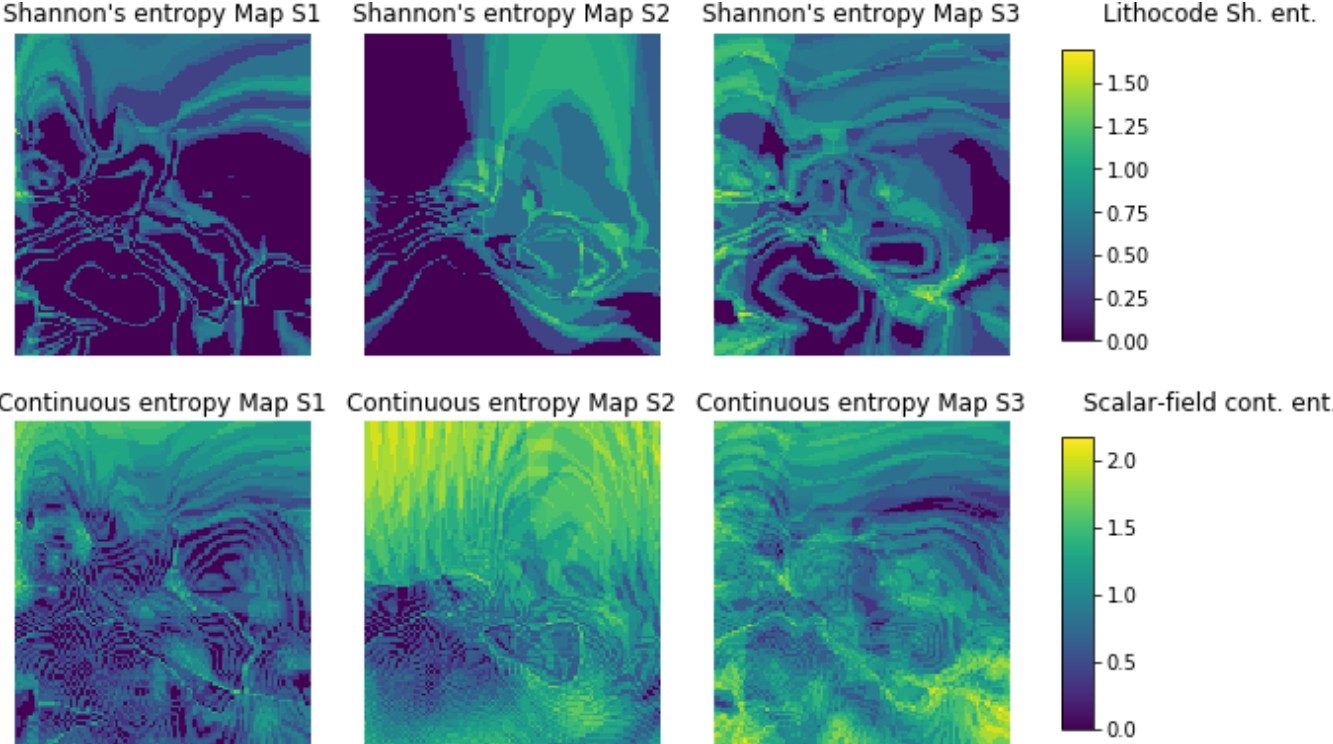

**Figure 7.** Horizontal sections entropy voxets; the first row shows Shannon's entropy over the ensemble of lithocode voxets, the second row displays the continuous entropy over the ensemble of scalar-field voxets; the left column refers to scenario 1, the middle column to scenario 2 and the right column to scenario 3.

It requires $P$ and $Q$ to share the same support $\mathcal{X}$ and can be computed for continuous or discrete variables. Here, we assume that for our pair of voxets $V_P$ and $V_Q$, the support of our random variables $P$ and $Q$ respectively, is discrete and of size $n$ (possibly $n$ bins for discretized continuous variables).

Denoting the support of $P$ and $Q$ by $(x_i)_{i=1...n}$, $p_i = \mathrm{Prob}(P = x_i)$, $q_i = \mathrm{Prob}(Q = x_i)$ and $m_i = \frac{p_i + q_i}{2}$, the Jensen-Shannon divergence is computed as in Eq. 2.

$$\mathrm{JSD}(P||Q) = \frac{1}{2} \sum_{i=1}^{n} p_i ln\left(\frac{p_i}{m_i}\right) + \frac{1}{2} \sum_{i=1}^{n} q_i ln\left(\frac{q_i}{m_i}\right) \tag{2}$$

### 3.4 Semi-variogram dissimilarity

Introduced by Matheron (1963), the semi-variogram measure the dissimilarity of values taken by random variables at different spatial locations as a function of a distance. Assuming stationarity and isotropy, it can be written as:

$\gamma(h) = \frac{1}{2} E\left[\left(Z(s) - Z(s+h)\right)^2\right]$, where $s$ denotes a spatial location, $h$ denotes a distance and $Z$ is the random variable of interest.





Using spatial samples of a random variable, it is then possible to compute an experimental or empirical semi-variogram over $n$
lags of width $\delta$ as follows:

$$\hat{\gamma}(h_i) = \frac{1}{2N_i} \sum_{(j,k)} |Z(s_j) - Z(s_k)|^2, \text{ where } h_i = (i - \frac{1}{2})\delta \text{ is the centre of the } i^{th} \text{ lag, } 1 \leq i \leq n \text{ and } N_i \text{ is the number of pairs } (j,k)$$

of points such that $(i-1)\delta \leq ||Z(s_j) - Z(s_k)|| \leq i$.

Given two empirical semi-variograms $\hat{\gamma}_1$ and $\hat{\gamma}_2$ (see e.g. Figure 8), we propose to use a weighted $l_p$ norm as defined in Eq.
3.

$$||\hat{\gamma}_1 - \hat{\gamma}_2||_p = \left( \frac{1}{\sum\limits_{1 \leq i \leq n} \frac{1}{h_i}} \sum\limits_{1 \leq i \leq n} \frac{1}{h_i} |\hat{\gamma}_1(h_i) - \hat{\gamma}_2(h_i)|^p \right)^{\frac{1}{p}}, \text{ where } p = 2 \text{ in the following examples.} \tag{3}$$

Note that the weight is inversely proportional to the lag distance, giving more importance to dissimilarities of the semi-
variogram for small distances. This allows to account for the structural noise and (dis-)continuity of the property fields.

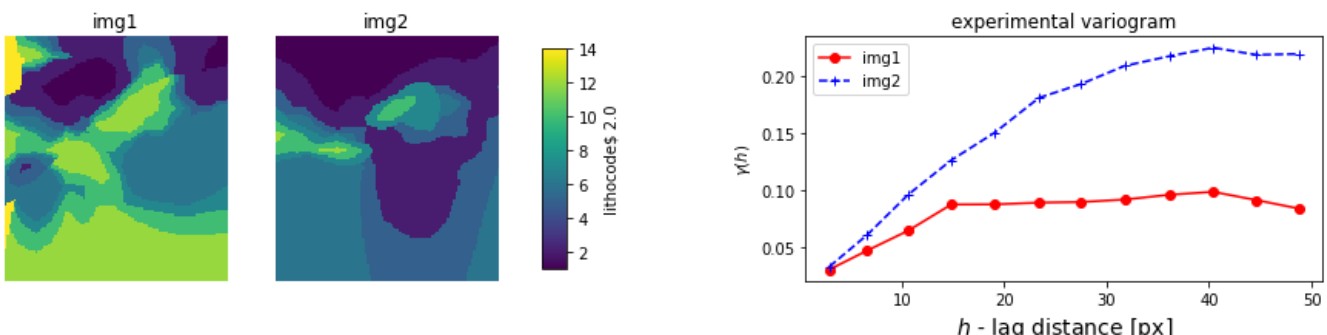

**Figure 8.** Example of experimental semi-variogram for two lithocode voxets, computed for the lithocode value of 2; the two first columns
show an horizontal section for each voxet; the last column shows the two corresponding experimental semi-variogram.

### 3.5 Connectivity dissimilarity

The existence of preferential flow-paths or barriers in the subsurface often has a strong impact in many geo-applications.
Their characterization can improve the management of groundwater quality, the extraction of geothermal energy, and help
mitigate the environmental impact related to either the production of non- and renewable resources from the subsurface or the
sequestration of carbon dioxide and waste (e.g nuclear waste). Renard and Allard (2013) have shown that connectivity cannot
be captured by topological indicators such as the Euler characteristic, nor by one-point or two-point statistics (e.g. by histogram
or semi-variogram analysis respectively). However, they have shown how a global percolation metric $\Gamma(p)$ and a lag-distance
connectivity function $\tau(h)$ are useful to characterize the connectivity of binary, categorical or continuous property fields.
Connectivity indicators have also been used in multiple-point statistics applications to characterize the quality of stochastic





simulations with respect to a training image (Meerschman et al., 2013) and in hydrogeophysical application for model selection (Pirot et al., 2019).

Let us consider a binary spatial variable $X \in 0, 1$, and a distance $h$. Then, the lag-distance connectivity function $\tau(h)$ is defined as the probability that two $h$-distant points $s$ and $s + h$ whose value of $X = 1$ are connected. For a binary voxel, two voxels are connected if a path through the face of successive neighbour voxels with the same property exists. The lag-distance connectivity function (see Figure 9) can be written as:

$$\tau(h) = Prob\left(s \xleftrightarrow{\text{connected}} s + h \mid X(s) = 1, X(s+h) = 1\right).$$

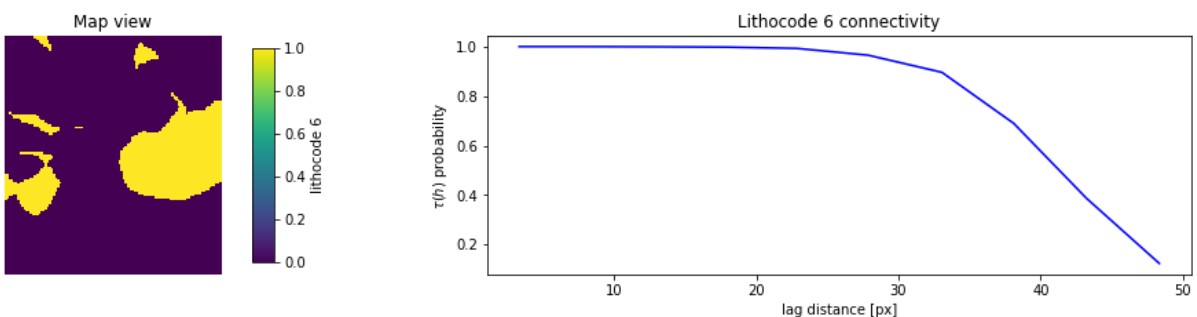

**Figure 9.** Illustration of the $\tau(h)$ connectivity function (right panel) computed on a 2D horizontal section from a binary 3D voxel (left panel).


Now let us assume that the percolation threshold $p$ produces a binary voxel characterized by the binary spatial variable $X \in 0, 1$. The global percolation metric $\Gamma(p)$ (see Figure 10) is the proportion of the pairs of voxels that are connected amongst all the pairs of voxels for which $X = 1$:

$$\Gamma(p) = \frac{1}{n_p^2} \sum_{i=1}^{N(X_p)} n_i^2 = \sum_{i=1}^{N(X_p)} p_i^2,$$

where $N(X_p)$ is the number of distinct connected components formed by voxels of value $X = 1$ and $p_i = n_i/n_p$ is the proportion of voxels forming the $i^{th}$ distinct connected component, $n_i$ being the size in voxels of the $i^{th}$ connected component and $n_p$ being the total number of voxels of value $X = 1$ in the voxel. Conversely, for the complementary set of voxels for which $X = 0$, we can compute $\Gamma(p)^c$ (see Figure 10). One can note that the two connectivity metrics are related as $\sum_h \tau(h) = n_p \Gamma(p)$ (Renard and Allard, 2013).

We propose two measures of connectivity dissimilarity between voxels, based either on $\tau(h)$ or $\Gamma(p)$ and $\Gamma(p)^c$. Let us denote by $N_c$ the number of considered class of values. Note that for a categorical variable voxel, the classes are defined by the category values, while for continuous variable voxel, they can be obtained by thresholding with $N_c$ percolation thresholds $p$. Let us consider $N_{lag}$ the number of lags (defined similarly as for the experimental semi-variogram in the previous subsection), $l_p = 2$ the distance norm and $\mathbb{1}_\tau = 1 - \mathbb{1}_\Gamma$ the indicator allowing to choose between $\tau$ or $\Gamma$ connectivity. The we can compute



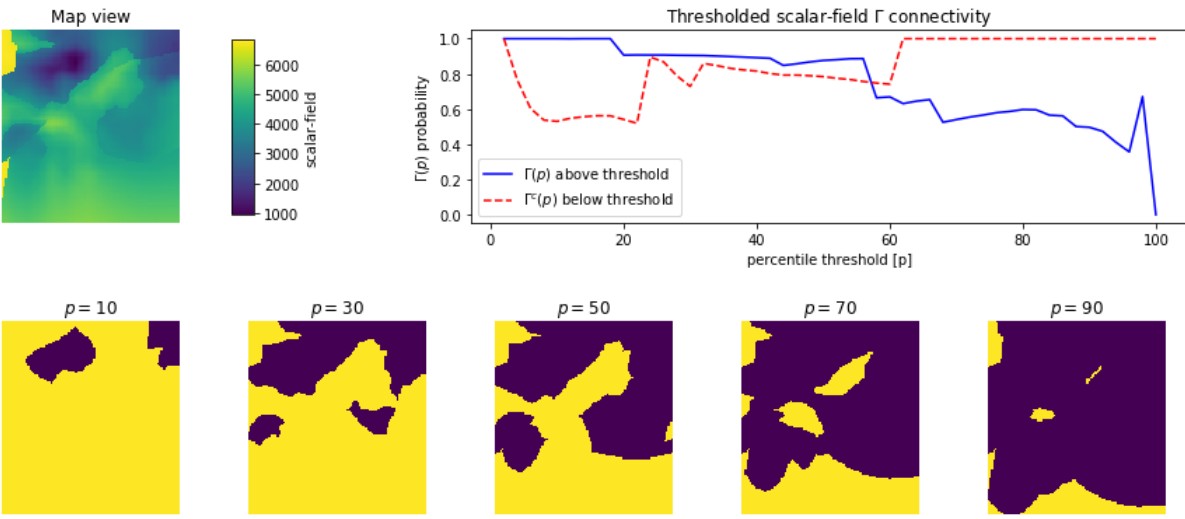

**Figure 10.** Illustration of a scalar-field voxet horizontal section (top left) and its global percolation metrics $\Gamma(p)$ and $\Gamma(p)^c$ (top right) computed over the 2D sections; the bottom row displays horizontal sections of binary fields obtained by applying different percolation threshold $p[\%]$ to the scalar-field 3D voxet; blue areas, above the percolation threshold, contribute to $\Gamma(p)$; yellow areas, below the percolation threshold, are used to compute $\Gamma(p)^c$.

the connectivity dissimilarity between two voxets as follows in Eq. 4:

$$d_{\text{CTY}}(\text{Voxet}_1, \text{Voxet}_2) = \sum_{i=1}^{N_c} \frac{1}{N_c} \mathbb{1}_\tau \left( \sum_{h=1}^{N_{lag}} \frac{|\tau_1(h) - \tau_2(h)|^{l_p}}{N_{lag}} \right) + (1 - \mathbb{1}_\tau)|\Gamma_1(i) - \Gamma_2(i)|^{l_p} \quad (4)$$

### 3.6 Multiple-point histogram dissimilarity

Multiple-point histograms (MPH Boisvert et al., 2010) are based on pattern recognition and have been primarily used in the field of geostatistics (Guardiano and Srivastava, 1993) to quantify the quality of multiple-point statistics simulations. Patterns are delimited by a search window whose dimensionality matches the one of the dataset. One can count unique patterns, however the number of unique patterns might be relatively important, in particular for continuous property fields. In that case, it might require to restrain the analysis to the most frequent patterns (Meerschman et al., 2013). An alternative is to base the analysis on pattern cluster representatives (see Figure 11). Here, using an $l_2$ norm distance between patterns and k-means clustering (Pedregosa et al., 2011, scikit-learn implementation), we classify all patterns into $N_c = 10$ clusters. Each cluster centroid or barycentre defines its representative.

In addition, voxets can be easily upscaled which allows MPH analysis of potentially large scale features with a small search window at high level of upscaling. Note that a given level $l$ of upscaling, the size of the dataset is divided by $2^l$ along each dimension. Here, to avoid property values smoothing, we perform a stochastic upscaling, i.e. in a 2D case, the upscaled value

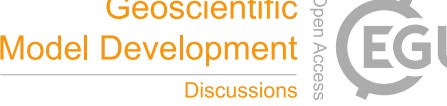

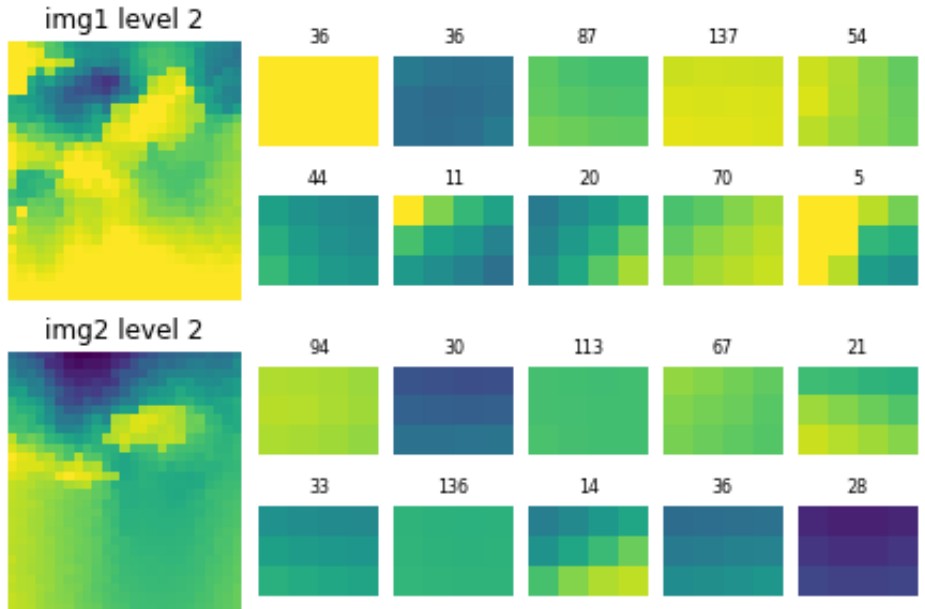

**Figure 11.** Illustration of multiple-point histogram cluster representatives and sizes for two scalar-field horizontal sections, at the $3^{rd}$ up-scaling level; the left column shows the two 2D voxets; for the other columns the first and second rows, respectively third and fourth rows, display the 10 cluster representatives and their size (number of counted patterns attached to the cluster representative) for the first voxet, respectively for the second voxet; their order reflect the best similarity between the cluster representatives for both voxets.

of a $2 \times 2$ subset of pixels is achieved by a uniform random draw among the values of the $4$ pixels. Cluster pattern identification

is performed at the initial resolution level ($l = 0$) and at all possible upscaled levels.

For a given upscaling level, once k-means clustering of patterns has been performed on two voxets or datasets, distances between cluster representatives of two images can be computed: $d(C_1^i, C_2^j) = \left( \sum_{w=1}^{N_w} (C_1^i(w) - C_2^j(w))^2 \right)^{\frac{1}{2}}$, where $C_1^i$ is the $i^{th}$ cluster representative for voxet 1, $C_2^j$ is the $j^{th}$ cluster representative for voxet 2 and $w$ denotes the index of the window-search elements.

The cluster representatives between two datasets are paired by similarity (smallest distance), and re-orderd such that $\forall i$, $1 \le i \le N_c$, $C_1^i$ is paired with $C_2^i$. To account for cluster size differences, the distance between paired cluster representative are weighted by proportion dissimilarities. It results in an MPH cluster based distance between voxets/datasets 1 & 2 defined as in Eq.5.

$$d_{\mathrm{MPH}}(\mathrm{Voxet}_1, \mathrm{Voxet}_2) = \sum_{i=1}^{N_c} \frac{1}{N_c} \left[ \left(1 + d(C_1^i, C_2^i)\right) \times \left(1 + \frac{|p_1^i - p_2^i|}{p_1^i + p_2^i}\right) - 1 \right], \tag{5}$$

where $p_1^i$ and $p_2^i$ are the proportions of the paired clusters $C_1^i$ and $C_2^i$ with respects to voxets 1 & 2 respectively.



One advantage of selecting cluster representative independently between two voxels is to lower computing requirements over large ensemble of voxels, performing the analysis for $N_v$ voxels instead of $\frac{N_v(N_v-1)}{2}$ pairs. However, performing k-means pattern clustering on two datasets might provide a more accurate and precise way to compute a distance between histograms with the same support of cluster representatives, allowing thus the use of Jensen-Shannon divergence for instance. One can

note that we accounted for the size of the clusters, but we could also consider the density spread or concentration around cluster representatives.

### 3.7 Wavelet decomposition coefficient dissimilarity

Wavelet decomposition is way to compress images. Each level of decomposition produces a series of coefficients. If computed for images to be compared, the dissimilarity of histogram of coefficients can be computed with the Jensen-Shannon divergence

(Eq. 2). Here, wavelet decomposition (Figure 12) is performed with the *PyWavelets* Python package (Lee et al., 2019) at all possible levels of decomposition and using the 'Haar' wavelet. Other wavelet could be used, however, tests have shown that such dissimilarity measures are note very sensitive to the choice of the wavelet (Pirot et al., 2019). Then a wavelet-based

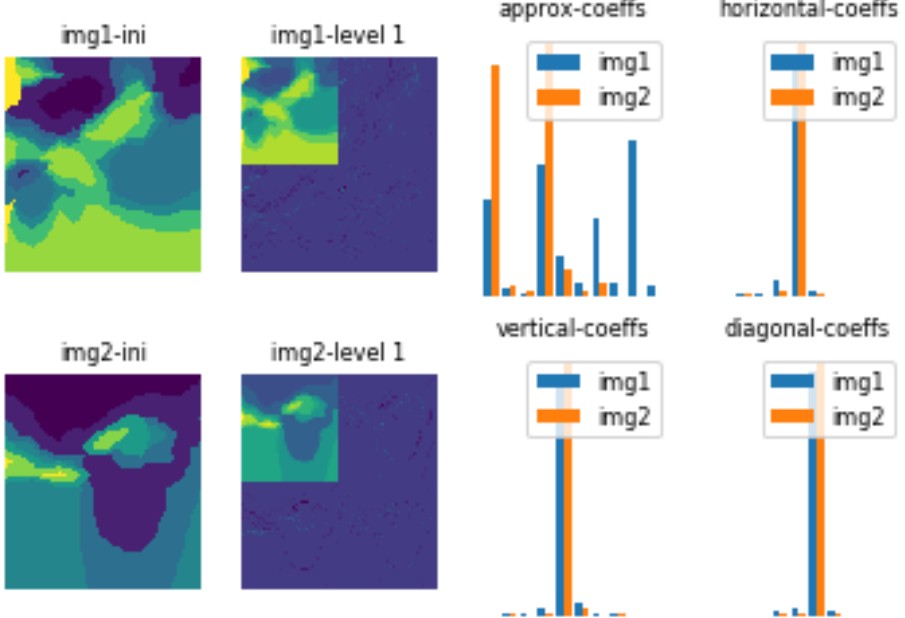

**Figure 12.** Illustration of a first level of haar-wavelet decomposition and the resulting coefficients histograms for two lithocode voxel horizontal sections.

dissimilarity measure between two voxel can be computed as in Eq. 6 by summing the Jensen-Shannon divergences computed





for all pairs of approximation and decomposition coefficients at all possible levels:

$$d_{\mathrm{WVT}}(\mathrm{Voxet}_1, \mathrm{Voxet}_2) = \sum_{i,j} \frac{JSD(C_1^{i,j} || C_2^{i,j})}{\sum\limits_{i,j} 1}, \tag{6}$$

where $C_1^{i,j}$ and $C_2^{i,j}$ denote the distributions of the $i^{th}$ coefficients at upscaling level $j$ for Voxet$_1$ and Voxet$_2$ respectively.

### 3.8 Topological dissimilarity

Thiele et al. (2016) give an overview of possible representations of the topology in the context of 3D geological modelling. Different levels of complexity (e.g. 1st or 2nd orders ...) can be used. Nonetheless, any topological indicator is a graph, that can take the form of an adjacency matrix. Thus to compute a topological distance between two 3D geological models (for instance as in Giraud et al., 2019), it seems natural to look at distances defined between graphs. Donnat and Holmes (2018) provide a comprehensive review of graph distances, used in the study of graph dynamics or temporal evolution. Though, here, in a geological context, we aim at comparing the topological diversity of an ensemble of geological models, we can use similar distances. Donnat and Holmes (2018) classify graph distances into three main categories as summarised below.

*Low-scale distances* capture local changes in the graph structure. The Hamming (structural) distance is the sum of absolute value of differences between two adjacency matrices, requires the same number of vertices (nodes) between the graphs - note that it is a specific case of the more general Graph Edit Distance. The Jaccard distance is defined as the difference between the union and intersection of two graphs. The Graph Edit Distance belongs to the NP-complete class of problems, and is not computed here. More info available in (Gao et al., 2010), at https://en.wikipedia.org/wiki/Graph_edit_distance, or in Part IV Chapter 15 of the Encyclopediae of Distances (Deza and Deza, 2009, p. 301). Note that some packages and implementations exist to compute the graph-edit distance, but have not been tested here ( GMatch4py, graphkit-learn, other proposed heuristic ).

*High level / spectral distances* are global measures and reflect the smoothness of the overall graph structure changes by measuring dissimilarities in the eigenvalues of the graph Laplacian or its adjacency matrix. Some examples are the IM distance (Ipsen and Mikhailov, 2003), $l_p$ distances on eigenvalues, or the Kruglov distance on eigenvector coordinates (Shimada et al., 2016).

*Meso-scale distances* are a compromise or combination of low-scale and spectral distances: Hamming-IM (HIM) combination, Heat-Wave distance, polynomial distance, neighbourhood level distances, connectivity-based distances.

Here, we propose to build first order adjacency matrices (see Figure 13) from 2D or 3D voxet models. For continuous property fields, the voxet is discretize in $N = 10$ classes of values defined by $N$ equi-percentile thresholds over the distribution of the combined voxets. We compute two topological distances: the structural Hamming distance and the Laplacian spectral distance (Shimada et al., 2016).

Note that graphs characterizing geological model topology could be defined as attributed graph, to contain more information (edges properties such as age constraint, type of contact; vertices properties such as formation type, geophysical properties). Thus more specific measures could be developed to take into account such characteristics. However, it would rely on the





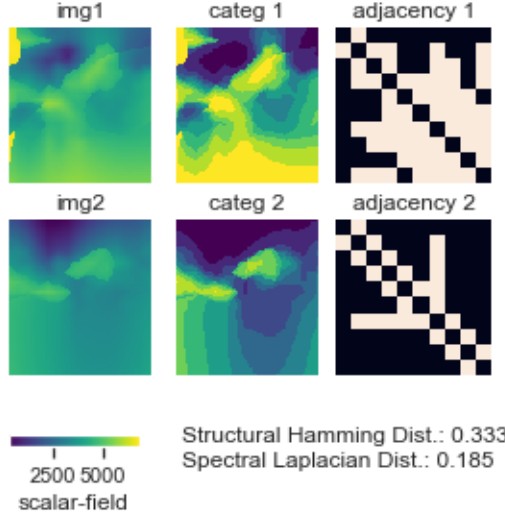

**Figure 13.** Illustration of topological distances and adjacency matrices in the right column for two categorised voxets in the middle column, derived from two scalar-field voxet horizontal sections in the left column.

ability of geomodelling engines to provide these topology graphs with each model, and there is no guarantee that it would be meaningful for the inference of geochronology from geophysics.

### 3.9 Results: indicator comparison

Local measures of uncertainty (see Figure 14 & 15) and global indicators (see Figure 17 & 18) have been computed for 2D, 315 3D, categorical and continuous variable voxets (lithocode, scalar-field) for an ensemble of 30 model-sets. We also provide a comparison of the required computing time for the different indicators and highlight the contributing factors/parameters (see Table 2).

One can see from Figure 14 that Shannon's entropy and cardinality computed from lithocode voxets can have a good correlation. However, equivalent indicators continuous entropy and averaged normalized range and standard deviation computed from 320 scalar-field voxet (Figure 15) have very little in common. Indeed, the standard deviation or the range of values are sensitive to extremely different values, while the continuous entropy is sensitive to the proportion of categories of values.

Figure 16 shows that histogram dissimilarities computed from lithocode or scalar-field voxets have a good correlation. Multi-dimensional scaling (MDS) plots reveal that dissimilarities are smallest within scenario 1 and then within scenario 3 while they are greatest within scenario 2. MDS plots show that scenarios 1 and 3 model-sets overlaps while scenario 2 model-sets are 325 characterized by less similar histograms and thus scenario 2 sample cloud of points form a distinct cluster.

Figure 17 shows some correlation between histogram, semi-variogram, wavelet and structural Hamming based measures from the lithocode voxets. Figure 18 shows a good correlation, between histogram dissimilarity, wavelet based dissimilarity and structural Hamming distance from the scalar-field voxets, but not as strong as when computed from the lithocode voxets.

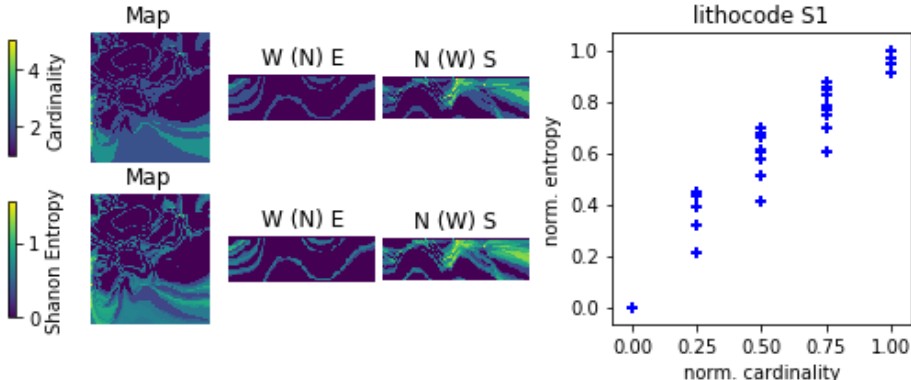

**Figure 14.** Comparison of local uncertainty measures for an ensemble of 10 lithocode 3D voxets for scenario 1; 3D visualisation looking from the NW of the voxet, the top surface of the voxet an EW section at the northern face of the model looking from the south, a NS section on the western face of the voxet looking from the east.

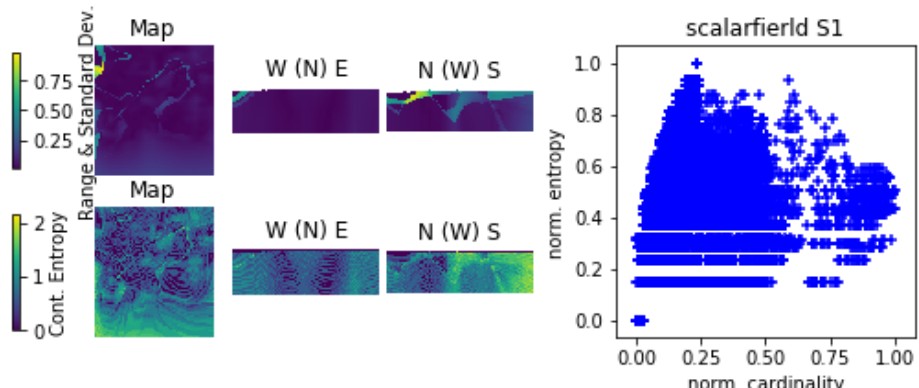

**Figure 15.** Comparison of local uncertainty measures for an ensemble of 10 scalar-field 3D voxets for scenario 1; 3D visualisation looking from the NW of the voxet, the top surface of the voxet an EW section at the northern face of the model looking from the south, a NS section on the western face of the voxet looking from the east.

## 4   Discussion

Several factors might explain why a majority of practitioners do not consider input data uncertainty, but all are related to the limited resources available to practitioners (knowledge, algorithms, computing time, project timeframe or funding). One of them is that data uncertainty is not quantified at the time of data acquisition or not available for some measurements, which is the case when only one measurement or observation of geological data is made at a given location (e.g. for azimuth, dip, and lithology). Reasonable metadata standards may help to enforce error quantification, or at the very least provide some informa-

tion about the nature of data collection as to infer where and what magnitude of error may be present, but even these are poorly





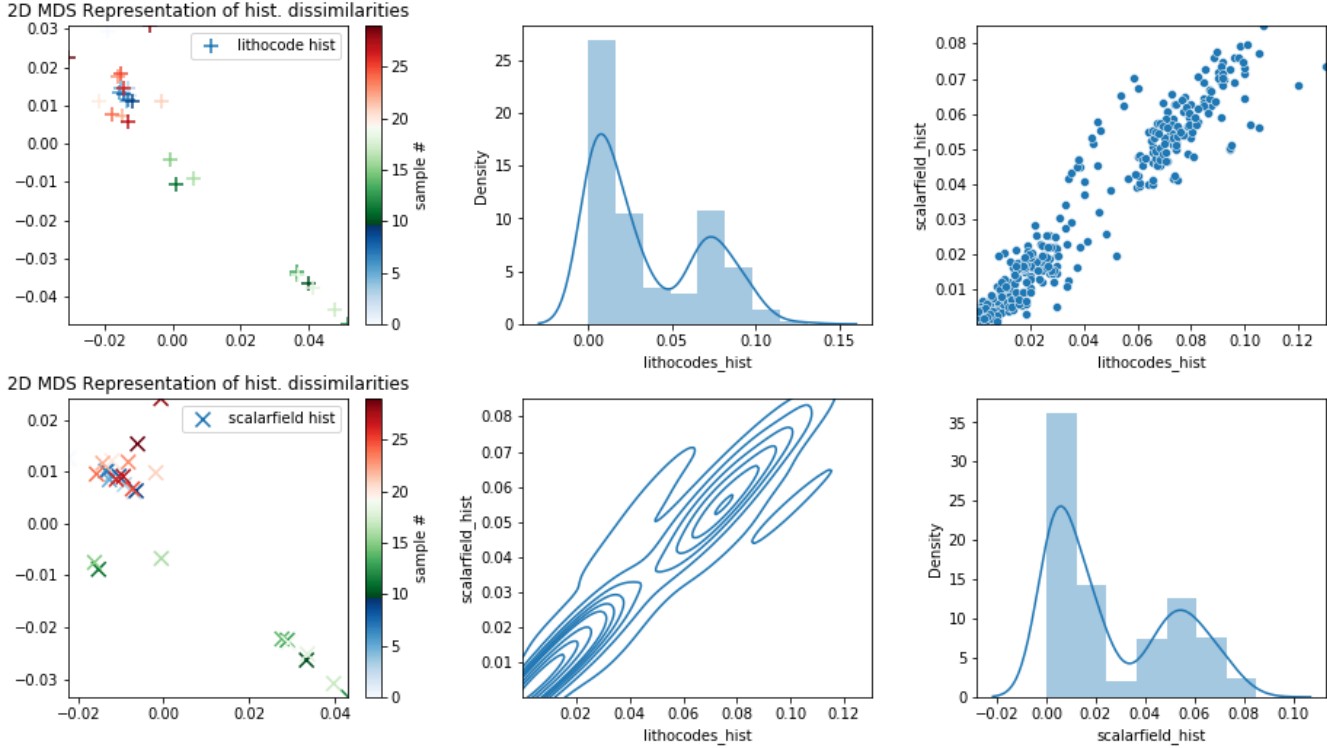

**Figure 16.** Comparison of model set histogram dissimilarities across the three scenarios; the left column displays a 2D MDS representation of histogram dissimilarities computed from lithocode voxets (top row) and from scalar-field voxets (bottom row); samples 0 to 9 belong to scenario 1, samples 10 to 19 belong to scenario 2 and samples 20 to 29 belong to scenario 3; the middle and right column subplots show histogram and density, joint density and cross-plot between histogram dissimilarities computed from lithocode voxets or from scalar-field voxets.

or not recorded. Although repeated independent measurements would provide uncertainty estimates, procedures and limited time or budget resources are a hindrance. Sometimes, knowing the survey setup and the instrumentation characteristics, such as their precision and accuracy might avoid repeating field measures and allow for an estimation of measurement uncertainty. Inversion algorithms used for geophysical data integration can also provide estimates of geophysical data errors.

A second reason is related to the required time and associated costs of modelling. Indeed, the process of data integration only uses a very limited amount of automation, thus the generation of a single model consumes already most of the practitioners' resources. In addition, the complexity of real world data often leads to a substantial number of parameters. Thus for high-dimensional problems, uncertainty propagation requires sufficiently large model ensembles to be representative, which might not be compatible with the limited resources available to the practitioners.

A third reason is due to the fact that assumptions, such as choices in geological interpretations, made during the modelling process are not always tracked. And when they are, they are often 'forgotten' at the next stage of the modelling workflow





**Table 2.** Complexity and computing time for local and global measures of uncertainty using a single Intel(R) Core(TM) i7-8550  1.80GHz, based on an ensemble size of $N = 10$ geological models.

| Measures | Number of evaluations | Total CPU time (HH:MM:SS) | influential parameters |
|---|---|---|---|
| Cardinality | 2 | 00:00:02 | $n_{voxels}, n_{voxets}$ |
| Entropy | 2 | 00:00:05 | $n_{voxels}, n_{voxets}, n_{bins}$ |
| Histogram dissimilarity | 60 | 00:00:38 | $n_{voxels}, n_{voxets}$ |
| Semi-variogram dissimilarity | 60 | 00:09:42 | $n_{voxels}, n_{voxets}, n_{categ}, rate_{sub-sampling}$ |
| Connectivity dissimilarity | 60 | 00:07:02 | $n_{voxels}, n_{voxets}, n_{categ}, rate_{sub-sampling}$ |
| Multiple-point histogram dissimilarity | 870 | 00:37:45 | $n_{voxels}, n_{voxets}, rate_{sub-sampling}$ |
| Wavelet decomposition coefficients dissimilarity | 870 | 00:00:38 | $n_{voxels}, n_{voxets}$ |
| Topological distances | 870 | 00:02:49 | $n_{voxels}, n_{voxets}, n_{bins}$ |

(Jessell et al., 2018). When these assumptions or justifications are recorded, they are described in metadata or in distinct reports. Consequently, conceptual uncertainty, which describes alternative yet plausible stratigraphy, tectonic and geodynamic settings is also ignored.

Another possible reason is that uncertainty is ignored out of convenience (Ferré, 2017) or by ignorance, lack of knowledge or education about the importance of uncertainty quantification. However, the formulation of the collected answers suggest that it is not the case for the surveyed practitioners, who rather acknowledge the importance and need for tools or algorithm to integrate uncertainty quantification in their modelling workflow.

While about 11% of the respondents indicate that they perform a quantitative uncertainty quantification, it is limited to
aleatoric uncertainty, i.e. data measurement errors. However, the lack of spatial data samples contribute to epistemic uncertainty and our limited contextual knowledge adds up to conceptual uncertainty. In addition, it is generally expected that these sources of uncertainty have a bigger impact on predictive uncertainty (Pirot et al., 2015). Thus, in addition to develop tools to facilitate aleatoric uncertainty quantification for practitioners, accessible tools integrating epistemic and conceptual uncertainty quantification need to be developed and promoted in the minerals industry.

Restricting the survey to a few open questions encouraged participants to take the survey and to express their uses, needs and opinion with limited perception bias, however it did not allow to quantify the gathered answers, and often requires some interpretation. Nevertheless, it is a first step in acknowledging the practices and needs of 3D geological modellers in the minerals industry. Another limitation of the survey is that it does not look at the practice and needs of other fields like the petroleum industry (Scheidt et al., 2018), geothermal industry (Chen et al., 2015) or hydrogeology (Pirot et al., 2019), though
they share similar scientific problems and also propose interesting solutions to deal with uncertainty quantification.





For model ensemble of size $n$, the calculation of local uncertainty indicators such as cardinality or entropy voxels seem to be much faster $O(n)$ to compute than global indicators $O(n^2)$, in particular when dealing with discrete or categorical variables. In addition, local indicators are most convenient to visualize uncertainty in 2D or 3D spaces. However, they cannot inform about the variability of important specific features such as connectivity or topology that can be estimated with global uncertainty

indicators. Thus, depending on modelling objectives and relevant features or characteristics, both local and global uncertainty indicator should be considered.

Presumably, the presented indicators are non-exhaustive and remain a subjective choice, even though all of them are already used rather individually in the geo-modelling community. Pellerin et al. (2015), for instance, propose other specific global geometric indicators. Here, we have focused on indicators that have shown some usefulness and practicality. Local indicators

could be extended to higher moments of the voxel-valued probability distributions, such as to consider asymmetry for instance. Global indicators could be extended to summary metrics of lower dimensional model representations, that could be obtained from discrete-cosine transform (e.g. Ahmed et al., 1974), (kernel-) principal component analysis (e.g. Schölkopf et al., 1997) or (kernel-) auto-encoding (e.g. Laforgue et al., 2019) for instance. This could be particularly appealing as it could reduce indicator computing costs drastically, but might not allow to identify the specific features of interest in a representation space

of lower dimensions.

Last but not least, one can note that we compared ensemble of models, whose underlying characteristics such as various lithological units derived from a shared stratigraphy and scalar-fields generated under similar assumptions are consistent. However, we must warn that the various indicators are compatible with differences in property ranges or meaning (for categorical variables), and thus it is the responsibility of the user to ensure the coherence of the model ensemble used as input for the

uncertainty computation.

## 5 Conclusions

The survey clearly shows that practitioners acknowledge the importance of uncertainty quantification; a majority recognize that they do not perform uncertainty quantification at all and all would like to do better. From this survey, we have identified four main factors preventing practitioners to perform uncertainty quantification: lack of data uncertainty quantification, com-

puting requirement to generate one model, poor tracking of assumptions and interpretations, relative complexity of uncertainty quantification. Here, as a first response, we have provided the geo-modelling community with **loopUI-0.1**, an open-source *Python* package to compute local and global uncertainty indicators. Then, to increase the confidence in predictions from 3D geological model, efforts should be made to explore conceptual uncertainty (Laurent and Grose, 2020), as well as towards the implementation of systematic geological data uncertainty quantification, and the exploration of parametric and epistemic

uncertainty (Pirot et al., 2020). It should be performed appropriately at all scales, across all geoscientific methods, such as the extraction of additional lithological data from drillhole databases (Joshi et al., 2021). To encourage uncertainty propagation among practitioners, accessible and compatible algorithms should be offered 1) to extract automatically geological data from open-databases (Jessell et al., 2021), 2) to quickly generate plausible geological models from a given dataset (Grose et al.,



2021) in interaction with geophysical data integration (Giraud et al., (this issue) and at an appropriate resolution (Scalzo et al.,

2021). This special issue (Ailleres, 2020) already provides elements of an answer to these problems and is expected to host future advances on these topics.

*Code and data availability.* The detailed survey questions are available in Appendix A and the gathered anonymized answers are available on request. The code to compute the uncertainty indicators and a set of illustrative notebooks are available at https://doi.org/10.5281/zenodo. 5656151.

**Appendix A: Survey**

For each model scale that you encounter in your work, please answer all the questions of the survey (1-10):

**A1  PART I - scale**

1. What are the scale and characteristic dimensions (width, length, depth and resolution) of the models that you build or use?



**Figure 17.** Comparison of global normalized dissimilarity measures for an ensemble of 30 lithocode 3D voxets across the three scenarios; cross-plots and density plots by pair of normalized dissimilarity measures; his - histogram, 2ps - semi-variogram, mph - multiple-point histogram, cty - connectivity, wvt - wavelet decomposition coefficients, shd - topological structural Hamming distance, lsg - topological Laplacian spectral distance.



**Figure 18.** Comparison of global normalized dissimilarity measures for an ensemble of 30 scalar-field 3D voxets across three scenarios; cross-plots and density plots by pair of normalized dissimilarity measures; his - histogram, 2ps - semi-variogram, mph - multiple-point histogram, cty - connectivity, wvt - wavelet decomposition coefficients, shd - topological structural Hamming distance, lsg - topological Laplacian spectral distance.





| Mine (<1 km dimension, ˜ 10 m resolution) |
| Brownfields (1-10 km dimension, ˜ 100 m resolution) |
| Greenfields/regional (>10 km dimension, ˜ 1 km resolution) |

2. What are the dimensions and grid cell resolution of your models?

| Model width (m) | |
| Model length (m) | |
| Model depth (m) | |
| Horizontal resolution (m) | |
| Vertical resolution (m) | |
| Main purpose (e.g. resource estimation) | |

## A2  PART II

Output

3. Which objectives do geological models help you to achieve?

4. How are they useful to fulfil other needs (and which ones)?

Input

5. What kind of input data, and what quantity and quality metrics (if any) are used to build your geological models?

Current modelling

6. How is the uncertainty of input data assessed and taken into account?

7. How is the geologist/modeller's interpretation recorded into the model (recorded tracks of assumptions, choices and
justifications)?

8. What is the usual modelling workflow and which tools or algorithms are involved?

9. How are performed data integration and upscaling? Which tools or algorithms are involved?



Improvements

10. What are the limitations of existing geological models to achieve your current and future objectives? How do you prioritize them and what kind of solution would you imagine?

*Author contributions.* The survey was initiated by Guillaume Pirot, developed jointly with Ranee Joshi and with the support of all other co-authors. All co-authors distributed the survey and collected answers. Answers were processed and analysed by Guillaume Pirot. The manuscript was mainly written by Guillaume Pirot, with contributions from all co-authors.

*Competing interests.* No competing interests are present.

*Acknowledgements.* This work is supported by the ARC-funded Loop: Enabling Stochastic 3D Geological Modelling consortia (LP170100985) and DECRA (DE190100431) and by the Mineral Exploration Cooperative Research Centre whose activities are funded by the Australian Government's Cooperative Research Centre Programme. This is MinEx CRC Document 2021/5.





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
