# Peer review of "loopUI-0.1: uncertainty indicators to support needs and practices in 3D geological modelling uncertainty quantification"

_Geoscientific Model Development, 2021_

## Author Comment (AC1)

Anonymous Referee #1, 25 Feb 2022

The paper is technically sound and generally well-written. It proposes and showcases tools for the assessment of geological models in the minerals industry, from the green-field scale to the mine scale, therefore it is of interest to the readership of Geoscientific Model Development.

> Answer: We thank the referee 1 for his/her review and useful comments.

I recommend publication subject to the following revisions:

Main comments:
1) Lines 30-31: when the authors refer to measurement errors, what about geological mapping or logging errors (due to the geologist's criterion, for example) that may not cancel out after replication of the measurement? Some comments on this issue would be welcome.

> Answer: When a measurement involves human criteria, such as the choice of specific locations for geological mapping or some interpretation in lithological classification or in the identification of boundaries, the sampling error can be characterized in repeating the mapping or logging by independent geologists. In other words, bringing in more geologists turns it into an expert elicitation exercise, which can be a valuable way of reducing uncertainty. The text has been updated with "repetitive independent sampling".

2) Line 197: isn't it too strong to assume "isotropy"? I believe that anisotropic variations are common in geological modelling

> Answer: The general formulation as given line 198 assumes anisotropy. $h$ could be a vector rather than a distance to deal with anisotropy. Directional semi-variograms are also a way to deal with anisotropy. Anisotropy will affect the shape of an omnidirectional semi-variogram. Here, to keep the dissimilarity measure simple, we assess the omnidirectional semi-variogram. Complementary details have been added in this section.

3) Lines 207-208: the experimental variogram is not always well-behaved at short distances, to the weighting may render the indicator in Eq. (3) highly sensitive to the short-distance behavior and nugget effect.

> Answer: This is true when dealing with sparse spatial data such as borehole or well data. Here, as we compare fully populated voxels, we are not concerned by this issue. A comment has been added for the reader in the corresponding section.

4) In addition to the presented indicators, would contact relationships between lithocodes (measured through transition probabilities, transiograms or cross-to-direct indicator variogram ratios) be worthy of interest? Again, some comments would be welcome.

> Answer: Indeed, these could be interesting topological indicators. As stated in the discussion, the proposed indicators are non-exhaustive and remain a subjective choice. To integrate this suggestion, the corresponding paragraph has been updated in the discussion.

Minor comments
1) There is a mix of US (e.g.: "minimize", "summarize") and UK ("summarises", "anonimised", "modelling") English

Answer: This has been corrected.

2) I am not familiar with the word "voxet" (seemingly, a set of voxels): this could be defined to avoid confusion

Answer: A definition has been added in the introduction.

3) Line 41: the date of the reference is 1978, not 1976

Answer: This has been corrected.

4) Line 45: there is a question mark before Sambridge

Answer: This has been corrected.

5) Lines 72-72: what/where are the Centre for Exploration Targeting, Loop researcher and related networks?

Answer: We have added a few urls to give more details: https://www.cet.edu.au/personnel/ , https://www.cet.edu.au/members/ , https://loop3d.github.io/loopers.html , https://www.linkedin.com/groups/6804787/members/ .

6) Figure 6: does the fourth row represent the "average normalized range and standard deviation", or the "average squared normalized range and variance" (the caption of the subfigures is not consistent with the figure caption)

Answer: This is the "average normalized range and standard deviation". The subfigure titles show a /2, not a square.

7) Line 167: can be computed

Answer: This has been corrected.

8) Lines 196-199: "s" is a vector, but is "h" a distance or a vector? Is Z a "random variable" or a "random field"? Notation should be revised for consistency

Answer: Z is a random field. $h$ is a vector in the more general case, but ban be viewed as a distance in the isotropic case. This has been clarified in the text.

9) Line 203: inside the norm, it should be s_j - s_k, rather than Z(s_j)- Z(s_k)

Answer: Thanks for spotting this mistake. This has been corrected.

10) Line 236: considered classes

Answer: This has been corrected.

11) Figure 10: the caption in the top right subfigure should be \Gamma(p)^c

Answer: This has been corrected.

12) Line 324: overlap

Answer: This has been corrected.

13) Lines 526 and 533: who are "et al."?

Answer: This has been corrected.

---

## Author Comment (AC2)

Anonymous Referee #2, 09 Apr 2022

Dear Editor

Determining the uncertainty in geological models is a crucial steps as it directly affects the next steps of geoscience projects.

In this Manuscript the authors have proposed some indicators for determining the uncertainty in the geological models. This is interesting and well written manuscript. I found this manuscript to be a very interesting read, specially from practical point of view, and one that is certainly worthy of publication.

Answer: We thank referee 2 for his/her positive review and comments.

I have just some small comments that I would like to be addressed by the authors:

Beside the indicators that you have already mentioned in manuscript, What other parameters one can used as an indicators (local or global) for determining uncertainty in geological models?

Answer: As stated in the discussion, the presented indicators are non-exhaustive and remain a subjective choice. One could also use through transition probabilities, transiograms or cross-to-direct indicator variogram ratios. Another possibility is to consider summary metrics of lower dimensional model representation.

You have proposed different uncertainty indicators. Which of proposed indicators would be preferred to use firstly? Which of them are more reliable and adequate for determining uncertainty?

Answer: We cannot recommend a specific indicator. Some indicators might be more suited in specific circumstances (specific to the modelling objectives). However, looking at the most shallow learning curve for a typical modeller/geologist… might suggest starting with cardinality, that simply states how many different lithologies are present at a given location. Then entropy will become more appropriate when the geologist starts to compare ensembles of models with different stratigraphies (in which case the total number of lithos will change, making cardinality inappropriate). The geologist may then want to know what effect all this uncertainty has on relationships expressed in the model, so they may then use the hamming or spectral distances. In addition, a good strategy might be to compute as many indicators as you can afford in your budget (see Table 2 for indicative computing requirements), another one would be to select a subset of voxels from your ensemble on which you can compute all indicators and perform a selection of the most informative or suitable ones prior to computing them on the whole set. The discussion has been completed accordingly.

Line 294: I don't consider Wikipedia as a scientific reference. Please eliminate it from you manuscript.

Answer: This has been corrected.

---

## Author Comment (AC3)

Anonymous Referee #3, 16 Apr 2022

Overall, I thought this was a very well written manuscript. Any tool(s) that encourage EDA, involve exploring uncertainty, and make this accessible are needed in our industry. My main concern is the difference between uncertainty quantification and visualization tools. The authors should make it clear that the focus here is EDA and visualization, while the uncertainty quantification (local or global) is not really the focus. Input data/parameter uncertainty (p19) is a good thing to focus on and is different to local/global uncertainty, which is not the focus here; for example, this usually requires some form of model, authors mention a 'Gaussian world'.

Answer: We thank referee 3 for his/her review and comments, contributing to the clarification of the manuscript. Assuming EDA = "Exploratory Data Analysis", the main objective of the paper is to provide tools that makes it easier to assess and visualize uncertainty on ensembles/sets of 3D models/parameters, based on quantitative criteria, as stated in the abstract. Exploratory Data Analysis is thus not the focus of the paper, however, it is critical step in the process of geological modelling, prior to generate ensemble of realizations. The proposed indicators are demonstrated on ensemble of non-Gaussian continuous and categorical models. This has been clarified in the introduction.

Major Changes:

I personally have not completed research based on surveys, but there are standard methods for presenting and developing these surveys. There are many details of surveys that are important for understanding/interpreting the results, some of these include: Which individuals (not names, but their background/industries/etc) were solicited to complete the survey? How many responded vs. how many were asked? Is 35 enough? What industries responded or were asked? Does this represent a reasonably diverse cross section through the industry or were only Australian professional in mining surveyed? Were these geomodelers, managers, junior/senior geologists/engineers? How was bias minimized? How were the questions decided upon? How were the multiple-choice answers selected to make sure the project goals were achieved? The majority of the justification in the work is based on conclusions drawn from these survey results, please expand on the details.

Answer: It was distributed among the 3D Interest Group (3DIG), Centre for Exploration Targeting (CET) members, Loop researcher and related networks. About 150 persons were given the opportunity to participate. The solicited participants are essentially based in Australia but from a number of nationalities, with interests in geological modelling for mining applications. They are either from the industry or academia, from junior to senior profiles. There were no multiple-choice questions. Most questions were open-ended to maximize our chance to learn about different uses and practices, as well as to minimize induced bias whenever possible. Added details have been updated in section 2.1.

I am not sure how Section 3 follows from the survey Section 2. There are many existing software/tools/scripts/etc available to quantify uncertainty and then visualize uncertainty (which I would consider different goals). Based on Q10, it seems like the biggest issue is underestimation of global uncertainty, but this is not addressed. Poor transition between sections 2 and 3.

Answer: Here we address in particular one of the needs identified in the survey: the lack of tools to quantify and visualize uncertainty among an ensemble of 3D voxets (prior or posterior

ensembles). It is of utmost importance for practitioners as it allows re-interpretations of data or scenario importance with respect to geological uncertainty or predictive uncertainty.  A transition between Section 2 and 3 has been added accordingly.

P8 Why are these 3 scenarios considered? Is this problem specific? Is this a good analysis for all datasets?

Answer: The different scenarios showcase how under-sampling, either due to inaccessible area (scenario 2) or lower density sampling (scenario 3), affects geological uncertainty in comparison to a more complete dataset (scenario 1).

I am concerned readers would confuse the models generated in Figure 5 with more typical continuous or categorical geomodels (i.e.. a Gaussian world). While there is certainly value to the proposed models, these would have limitations compared to industry best practice, please make this clear.

Answer: The technique used to generate the models does not matter here. The proposed models are here only to illustrate how the indicators work on ensemble of  continuous or categorical 3D voxets. The choice of modelling engine is up to the modeller and should comply with their objectives.

Not quite sure what is meant by 'underlying scalar field derived from implicit modeling'. Implicit modeling is a class of techniques, seems like something specific is used in Figure 5. Explain what is meant by this.

Answer: In the text, we refer the reader to Grose et al. (2021).  A geological model can be characterized explicitly (i.e by hand drawing) by a set of 3D surfaces that delineate boundaries between geological features. However, an explicit representation is very time consuming and not prone to the integration of additional data or knowledge. Implicit geological surface modelling rely the definition of a continuous scalarfield, whose selected isovalues define 3D interfaces ; the scalar-field is obtained by spatial interpolation of  identified values (e.g. lithological contact), and can be easily re-estimated when adding new data. More details can be found in

Lajaunie, C.; Courrioux, G. & Manuel, L., Foliation fields and 3D cartography in geology: principles of a method based on potential interpolation, *Mathematical Geology, Springer,* **1997***, 29*, 571-584

Or

Calcagno, P.; Chilès, J.-P.; Courrioux, G. & Guillen, A., Geological modelling from field data and geological knowledge: Part I. Modelling method coupling 3D potential-field interpolation and geological rules, *Physics of the Earth and Planetary Interiors, Elsevier,* **2008***, 171*, 147-157

Or

Guillen, A.; Calcagno, P.; Courrioux, G.; Joly, A. & Ledru, P., Geological modelling from field data and geological knowledge: Part II. Modelling validation using gravity and magnetic data inversion, *Physics of the Earth and Planetary Interiors, Elsevier,* **2008***, 171*, 158-169

P12 Why are you discussing connectivity? Is this for a mining application (the survey was mining focused) or petroleum or hydrology? Would help to know what groups was solicited for survey results and the target audience for your tools.

Answer:  As stated in section 3.5, the existence of preferential flow-paths or barriers in the subsurface often has a strong impact in many geo-applications. Their characterization can improve the management of groundwater quality, the extraction of geothermal energy, and help mitigate the environmental impact related to either the production of non- and renewable resources from the subsurface or the sequestration of carbon dioxide and waste (e.g nuclear waste). For this reason, we remain general.

Some minor comments:

Figure 2: I think the authors mean realization (not 'real')

Answer: This is right and the figure has been updated accordingly. TO DO IN THE CAPTION.

Figure 2: replace (or add) the type of data for 'data 1' and 'data 2' (I'm guessing drillhole/well samples and remote/production data).

Answer: It could be. More generally, data 1 is used only to build a prior ensemble of models; data 2 is used to reduce uncertainty and sample the posterior ensemble of models, which is by exploring the prior ensemble of models and selecting those who are more likely to reproduce/simulate data 2 within a level of error. The figure has been updated accordingly. TO DO IN THE CAPTION.

Figure 8: is this the variogram for the 'model'? Elaborate on the appropriateness of this. Clearly this is not the variogram for the underlying sample data.

Answer: These are experimental semi-variograms for two different 3D voxets of a binary variable (specific lithological code), as detailed in the caption.